# The Development of Equipment to Measure Mesh Erosion of Soft Tissue

**DOI:** 10.3390/ma14040941

**Published:** 2021-02-17

**Authors:** Amanda Schmidt, Gordon O’Brien, David Taylor

**Affiliations:** Department of Mechanical, Manufacturing and Biomedical Engineering, Trinity Centre for Biomedical Engineering, Trinity College Dublin, D02 PN40 Dublin, Ireland; amschmid@tcd.ie (A.S.); gordon.obrien@tcd.ie (G.O.)

**Keywords:** mesh, polypropylene, soft tissue, erosion, cutting, wear, creep

## Abstract

Mesh erosion is a phenomenon whereby soft tissue becomes damaged as a result of contact with implants made from surgical mesh, a fabric-like material consisting of fibers of polypropylene or other polymers. This paper describes the design and construction of a testing machine to generate mesh erosion in vitro. A sample of mesh in the form of a 10 mm wide tape is pressed against soft tissue (porcine muscle) with a given force, and a given reciprocating movement is applied between the mesh and the tissue. To demonstrate the capabilities of the equipment, we measured erosion using the same mesh and tissue type, varying the applied force and the reciprocating stroke length, including zero strokes (i.e., static loading). For comparison, we also tested four other samples of polypropylene with different edge characteristics. Analysis of the results suggests the existence of three different erosion mechanisms: cutting, wear and creep. It is concluded that the equipment provides a useful and realistic simulation of mesh erosion, a phenomenon that is of great clinical significance and merits further study.

## 1. Introduction

Surgical mesh is a fabric-like material made in sheet form by combining polymer fibers. The most commonly used type is made by knitting fibers of polypropylene (PP): several different commercial products are made in this way, including Prolene (Ethicon Corp, Somerville, NJ, USA) and Sutulene (Sutumed Corp, Fort Myers, FL, USA): Figure 1 shows examples.

Some meshes use different polymers, including degradable materials and those derived from biological sources, and some are made using other methods, such a weaving. They are widely used to make implantable devices for the repair and support of soft tissues and organs, such as hernia repair [3,4] and restraint of prolapsed organs in the pelvic area such as the vagina and rectum [5,6]. Another application, which became very popular over the last twenty years, is the use of mesh in the form of a tape of width approximately 10 mm to apply pressure to the urethra to prevent urinary incontinence [7,8]. This type of operation has been performed in very large numbers; for example, by 2017, it was estimated by the NHS in the United Kingdom that over 100,000 women in England had received mesh in operations for pelvic organ prolapse (POP) or stress urinary incontinence (SUI) [9].

In recent years, several major complications have been identified with these products, which have led to many countries imposing bans or restrictions on their use for the treatment of POP and SUI. One of these complications is mesh erosion of soft tissue. The edge of the mesh presses on an organ such as the urethra or vagina and moves back and forth during everyday activities such as running, sneezing or sexual intercourse. Brandao et al. developed a computer simulation, which predicted relative movements between mesh and tissue of the order of millimeters and forces of the order of Newtons [10], though it is fair to say that currently, there is a lack of detailed understanding of the biomechanics involved. The result is that the mesh tape can damage the tissue, causing pain and discomfort. Sometimes the mesh cuts completely through the organ wall, emerging inside the vagina, rectum or bladder, for instance [2,11]. Figure 1 shows an example of a piece of exposed mesh, which was surgically removed as a result. Such incidents cause considerable pain and distress to the patient and may compromise the intended function of the product. Analysis of clinical data suggests that mesh erosion occurs in approximately 3% of stress urinary incontinence devices and 10% of devices used for pelvic organ prolapse [7,12]. Erosion can occur very quickly, within days of the operation, but can also progress much more slowly, only becoming evident after several years.

Despite the importance of mesh erosion, our search of the literature up to 2019 revealed no previous attempts to reproduce and measure the phenomenon in vitro. Normally, when designing a new medical device, laboratory experiments would be conducted to optimize its functionality and to identify and study any possible complications. These experiments would normally begin by testing samples of the material and prototypes of the product, using animal organs ex vivo in place of the appropriate human organs. Such experiments may not accurately simulate all aspects for various reasons. For example, a piece of dead tissue may have different mechanical properties and will not be capable of self-repair and other biological responses. However, experience has shown that in vitro experiments can be a very useful first step to identify possible problems before proceeding to animal trials and eventually human clinical trials.

In the case of mesh products used for SUI and POP, we found publications reporting mechanical property data [13,14] and a few animal studies (e.g., [15]). We also found a large number of clinical case reports describing mesh erosion and other complications (e.g., [2,11]) and summaries of complications from large-cohort follow-up studies and meta-analyses [7,8,9]. However, as far as the open published literature is concerned, there appeared to be no in vitro studies of mesh erosion, so in 2019 we embarked on a program of work to develop equipment for this purpose.

Initially, we created a simple hand-held apparatus in which a length of mesh tape was held inside a hollow metal tube and moved back and forth in a sawing action to cut through a sample of porcine muscle held in a second tube [16]. This apparatus was sufficient to demonstrate the principle and to show that the rate of erosion varied with the applied force and the orientation of the muscle fibers. However, being operated by hand was subject to uncertainties with regard to the exact forces and motions being applied and was not suitable for long-term testing.

We subsequently developed another piece of equipment, which operates on the same concept, but affords better control and automation of the experiment. Recently, some results were published from this equipment, comparing different commercial mesh tapes, all tested at the same applied force [17]. Further results are available in a master’s thesis written by one of the authors [18].

In the present paper, we describe the equipment in detail, providing an explanation of its development and its key components. We also present new results to demonstrate the capabilities of the equipment to test at a range of applied forces and stroke lengths. We show how the analysis of these results provides insights into the mechanisms by which surgical mesh causes damage to soft tissues.

## 2. Methods and Materials

The original manually controlled equipment [16] (Taylor and Barton, 2020) is shown in Figure 2.

A tissue holder made from a piece of box-section steel with two 1 mm slots in the sidewall was held in a vice. The soft tissue sample (a cube of porcine muscle of size 8 × 8 × 8 mm from a butcher) was placed into the holder and supported from below. The second piece of box-section steel of larger size (25 × 25 mm) with side cuts was used to hold the mesh specimen (a tape of size 10 × 30 mm) and was moved back and forth to replicate a sawing motion. The downwards force of the mesh on the tissue was provided by the weight of the holder itself. Although this initial test provided repeatable results, an automated testing machine was required for longer experiments and greater accuracy.

The following sections describe the design requirements for the new equipment and how they were achieved, highlighting key components.

### 2.1. Automated Motion with Variable Stroke Length and Speed

The stroke length of the Taylor and Barton device was visually guided by the internal and external edges of the large and small pieces of box-section. The stroke speed was determined manually by the user.

For the automated device, a stepper motor and controller were selected based on expected test duration and frequency. The linear motion was achieved using a simple slider-crank mechanism, the amplitude of which can be varied using crank wheels with different offsets (see Figure 3). Test speed (i.e., the frequency of the reciprocating action) is controlled by the speed of the motor.

### 2.2. Stationary Mesh Holder; Frictionless Vertical Motion

A significant design change was to achieve the relative motion by keeping the mesh specimen in a fixed position and applying the reciprocating action to the tissue sample holder: this avoided having to manage the motion of the dead weights, which provided the applied load. An important design requirement was that the mesh and its holder would be able to move downwards freely as the tissue eroded. A longer piece of mesh tape (10 mm × 60 mm) was chosen to allow a greater range of reciprocating strokes. The holder was provided with screwed clamps to allow the tissue to be secured with a small amount of tension (10% strain). The tape is oriented vertically with one long edge pressing into the tissue with a load applied via a static weight. This orientation was chosen as a worst-case scenario in which all force is applied through one edge. This scenario seems to correspond to clinically observed erosion events. To ensure a free downward motion, a Perspex mesh guide was included with a sliding fit to act as near-frictionless support for the mesh holder while preventing sideways movement. The holder itself has a weight that provides a load of 0.3 N force: additional weights can be added (see Figure 4).

### 2.3. Moving Tissue Sample; Sample Hydration

Guide rails were provided to ensure the linear motion of the tissue sample holder. The sample holder was 3D printed from PLA and fixed in a water bath to ensure continued hydration during long-term tests. The tissue holder accommodates a 10 mm cube of tissue. The size of this sample may have an effect on the results, so we built in an allowance for changing the size of the holder. As in the original design, the tissue holder was provided with 1 mm wide vertical slots to accommodate the mesh during erosion. The water bath was machined from transparent Perspex to allow observation during testing. The bath was fixed to a low friction (Ertacetal C) base that was connected to the crank mechanism con-rod (see Figure 5). The soft tissue used was porcine muscle (“pork loin”) obtained from a butcher, placed so that the orientation of the mesh edge was perpendicular to the muscle fibers. Other types of tissue could be used, and the holder could be modified to take samples of organs such as the vagina or urethra.

### 2.4. Moving Parts Enclosed

The unit was fully enclosed for safety reasons. The enclosure, made from 12 mm polycarbonate, also incorporated support for the mesh clamp guide, using a rectangular hole with a stepped pocket in the upper surface. Figure 6 shows the complete equipment. Table 1 shows a summary of its technical capabilities.

## 3. Test Program

A test program was designed to investigate the capabilities of the equipment. The surgical mesh used was Sutulene (Sutumed Corp, Fort Myers, FL, USA), which we had already used in the earlier work [16,17]. The material, shown in Figure 7, was provided in the form of sheets of size 300 × 300 mm. It consists of knitted polypropylene fibers of a diameter of 150 μm. The knitting pattern creates cells of size 1.3 × 1.8 mm and confers significant anisotropy: we chose to cut the tape samples with their long axis parallel to the stiffest direction in the mesh. Some samples were cut using a scalpel, while others were cut using a CO_2_ laser (BRM 90130, BRM Lasers, Winterswijk, The Netherlands). This produced edges with different appearances (see Figure 7): the laser-cut edge showed evidence of local melting and fusion of fibers creating an edge that was more rounded in form but also more rigid. Commercial SUI and POP products are made using various manufacturing methods, some of which involve mechanical cutting, while others use lasers or other methods to generate local melting.

For comparison purposes, we also tested a polypropylene suture (size 6/0) to provide a smooth edge, along with three edges created using sheets of polypropylene of thickness 0.45 mm, which is similar to the thickness of the mesh material. One sheet was cut mechanically to create a smooth edge, while a second sheet was cut into a regular zig-zag pattern, creating serrations of depth 0.8 mm, which is similar to the roughness of the mesh edge (0.8 mm for the scalpel cut and 0.9 mm for the laser-cut edges). A third sheet was cut with serrations of greater depth: 2.5 mm. From the literature (Hong et al. 1998), we estimated the roughness of the suture and smooth sheet to be 0.01 mm.

Tests were carried out with four different applied loads: 0.3 N, 0.8 N, 1.3 N and 1.8 N. Two different stroke lengths were used: 20 mm and 2 mm. These values fall within the range of predicted forces and movements in vivo (10). A constant frequency of 1 Hz was used for the reciprocating motion. In addition, some tests were carried out with no reciprocation, applying a static load. The testing times varied from less than one minute to 48 h. The water bath was used to maintain hydration of the tissue in long-duration tests. After each test, erosion was quantified by measuring the depth of the cut formed in the tissue sample (see Figure 7) using digital calipers. The erosion rate was defined as the depth of the cut (in millimeters) divided by the total sliding distance (i.e., the length of the stroke multiplied by the number of strokes) in meters. Two further erosion parameters were also calculated: (a) the “erosion factor”, which is the erosion rate divided by the applied force, and: (b) the “time-based erosion factor”, which is the erosion factor, but dividing by time rather than sliding distance.

Statistical analysis was conducted using ANOVA (analysis of variance) to determine overall statistical significance and Student’s *t*-test to determine *p* values between groups, with a critical p value of 0.05. Microsoft Excel software was used.

## 4. Results

In total, 217 tests were conducted, in 43 different groups, covering combinations of four different levels of applied load and three different stroke lengths. At least five samples were tested for each load/stroke combination. Table 2, Table 3 and Table 4 show the average erosion rate in units of mm/m (or, in the case of the static testing, mm/day) for the various test groups.

Reproducibility was found to be good: for most groups, the standard deviation was less than 20% of the mean, only rising higher in 9 of the 43 groups, mostly in cases when the erosion rate itself was very low. The reliability of the data was sufficient to allow differences to be detected with statistical significance (*p* < 0.05). This showed, for example, that the mesh with a laser-cut edge had a significantly higher erosion rate than the mesh with a scalpel cut edge, and this was true for all four applied loads. Likewise, all specimens showed significantly different erosion rates when tested with different loads or different strokes, the only exceptions being the smooth PP sheet and suture, for which the increase in erosion rate with load was not significant at all loads, though the overall trend was still significant.

Figure 8 shows erosion rate plotted as a function of applied force for tests conducted with a stroke of 20 mm: the data are shown using two graphs here with different scales to better present the full range of erosion rates. Figure 9 shows the results for the smaller stroke of 2 mm. A simple linear relationship between force and erosion rate describes the data reasonably well for each of the groups, with R-squared values ranging from 0.6 to 0.9.

## 5. Discussion

These results demonstrate that the equipment is capable of generating reproducible data with relatively little scatter. One possible source of variability is the difference in the mechanical properties of the tissue from one sample to another. Other sources relate to operations that are carried out by hand, especially the cutting of the tissue sample to the correct size and the stretching of the mesh tape during attachment to the holder. Previously [16], we showed that if the direction of cutting (i.e., the longitudinal axis of the tape) coincides with the orientation of muscle fibers, then the erosion rate is much higher. In the present work, this was avoided by always cutting across the muscle fibers, and variability was reduced by always using the same muscle group. Examination of previous work on muscle properties [19] would suggest that, even so, variations in strength will occur from sample-to-sample. Testing a sample with greater strength is equivalent to applying a lower force, so this will have a proportional effect on the erosion/load relationship.

Given the generally linear dependence of erosion on load, it is convenient to define an overall erosion rate for each group (i.e., each combination of edge type and stroke) by dividing the erosion rate by force in Newtons, giving a factor with units of mm/m/N. We will refer to this as the “erosion factor” since it is analogous to the wear factor k, commonly used in tribological studies. The only difference is that in our case, the erosion is measured in terms of depth of cut, while in wear studies, it is normal to measure the total volume of material removed. The erosion factor is expressed in terms of the sliding distance, in meters, which is useful to allow us to compare results from different materials, edges and stroke lengths. However, it does not allow comparison with the static-load (creep) tests. Therefore, we also define a “time-based erosion factor” with units of mm/s/N.

Figure 10 shows plots of erosion factor and time-based erosion factor, as a function of edge roughness R.

Here the erosion factor has been averaged over all applied loads, which is reasonable given the linear relationships demonstrated above. The use of logarithmic scales allows us to obtain an overall view of trends in the results. It is evident that these trends are different for the three different testing conditions of 20 mm stroke, 2 mm stroke and static loading. For the 20 mm stroke, there is a strong effect, which is close to a simple linear relationship in which the rate of erosion is proportional to the roughness of the edge. A best-fit line drawn assuming a power-law relationship gives an exponent of 0.834. For the 2 mm stroke, however, the effect is weaker: erosion increases only by a factor of 3.0 when roughness increases by a factor of 250. Finally, for the static loading, there appears to be no effect of roughness on erosion or even a possible negative effect.

This suggests that different mechanisms are operating to cause damage and erosion. For the 20 mm stroke, the strong linear relationship suggests a cutting mechanism. In fact, a reasonable prediction of the data can be obtained using a very simple model in which it is assumed that the amount removed per stroke is equal to the roughness R. Thus, the erosion rate is simply given by 50 R since 50 strokes are required to achieve an overall travel distance of one meter. This model assumes that the edge of the specimen is pressed into the tissue by the applied force, to the entire depth of the roughness R, and that all the intervening tissue is removed during the stroke. This very simple model provides a reasonable fit to the data, as shown in Figure 10a. It should be noted that the applied force is not included in this model, so placing the line on this graph implies a force of 1 N. The model could be extended to include the effect of force, proposing that lower forces cause only partial embedding of the edge in the tissue, reducing the depth of cut and that at higher forces the cutting edge descends during the stroke.

For the 2 mm stroke, the above model is not able to predict the results: in fact, it would predict a rate of erosion ten times larger than that for the 20 mm stroke, when in fact, the erosion is much smaller. A clue to understanding this difference is the fact that the stroke length is now of the same order of magnitude as the roughness itself, at least in all cases except the very low roughness of the smooth PP sheet and suture. In addition, it was observed that during the stroke, the surface of the tissue moved back and forth somewhat with the specimen edge. In some cases, this was due to a slight mismatch in which the tissue sample was cut to be somewhat smaller than the tissue holder, leaving it loose. However, even when this was avoided, the tissue was seen to move with the sample edge by a distance of the order of 1 mm, due to friction between the specimen’s edge and the surface of the tissue. Though this was not precisely measured, it implies that the relative movement between tissue and the edge was about 1 mm, rather than the full stroke of 2 mm. It is likely that such a movement would not be sufficient to fully cut the portions of tissue lying between the arms of the roughness in the edges of the mesh and PP sheets. In these circumstances, a different mechanism may operate, more similar to wear than cutting. Small local movements will cause particles of tissue to be removed by abrasive wear, to the degree that is less dependent on the overall roughness R and more dependent on the local surface roughness of the material, i.e., the cut or extruded surface of the polypropylene itself. This may explain why the erosion factor was low for the smooth PP sheet and increased only moderately in the rougher edges.

Finally, another different mechanism may apply in the case of static loading. Since there is now no relative movement between the tissue and the PP, deformation and fracture of the tissue will occur by a creep mechanism. This will be affected by the local downward pressure of the PP on the tissue, which may explain why it was somewhat lower for the two mesh specimens compared to the smooth PP sheet due to the greater surface area of the polymer in contact with the tissue.

The very large differences in the erosion factors between the 20 mm and 2 mm strokes suggest that there will be little interaction between the cutting mechanism and the wear mechanism. If cutting is occurring, it will dominate over wear. The static loading results appear lower on the time-based erosion factor (Figure 10b), but this is somewhat misleading because in calculating this factor, we assume that the reciprocating motion is occurring continuously at a frequency of 1 Hz. In practice, it will be associated with occasional activities such as running, coughing, etc., while the creep mechanism, though smaller, will be operating continuously as a result of pressure exerted by the mesh on the organ.

Overall, our equipment appears to provide a reasonable simulation of mesh erosion, which is comparable to that observed clinically. As noted above, there is as yet limited information about the actual loads and movements, which occur in vivo. However, a computer simulation by Brandao et al. [10] predicted peak forces between mesh and tissue in the range of 1.6–3.4 N and relative movements of 6–15 mm, which are within the range of values used in our tests. A typical value of erosion rate for the mesh if erosion is proceeding by the cutting mechanism is 30 mm/m/N. Assuming an applied load of 2 N, and assuming that the mesh moves a total relative distance of 10 mm per day (which is likely to be an underestimate), then the mesh will be able to erode a depth of 10 mm (sufficient to cut through an organ) in a time of 17 days. This would correspond to the experiences of some patients for whom mesh exposure occurred within days or weeks of the operation. Considering, instead of the wear mechanism, for which a typical erosion rate would be 0.2 mm/m/N, the time to cause exposure increases to seven years, which corresponds to the experiences of other patients for whom the device functioned well for several years before causing complications.

In vivo, several other factors will be involved, especially the formation of scar tissue on the mesh, which tends to protect organs but also gives rise to contractile forces that can increase the applied load. Other factors relate to the placement of the device by the surgeon: some applied load is required for the device to function, but this is difficult to judge accurately at the time of placement. It is evident that the equipment developed in the present work will never be sufficient to investigate all aspects of the problem, but it can provide a useful first step, especially to study certain factors such as the nature of the mesh material and the method of formation of the edge. For example, the results are shown here (Table 2) revealed that a laser-cut edge causes considerably faster erosion than a scalpel-cut edge, even though they have similar roughness values. We obtained similar results for other mesh products, which have been published recently elsewhere [17]. This effect may be explained by the fact that the laser-cut edges are more rigid and less likely to deform during the cutting process. The use of equipment such as ours may help to understand the various factors that affect erosion and to prevent mistakes being made during the development of new products for SUI and POP.

## 6. Conclusions

The test equipment developed is capable of generating reproducible results with an acceptable degree of scattering, which allows one to investigate the effect of testing variables such as applied force and stroke length, as well as product variables such as edge roughness the method of manufacturing the edge;For a large stroke length of 20 mm, a cutting mechanism operates, and there is a strong effect of applied load and edge roughness on the rate of erosion;For a smaller stroke length (nominally 2 mm but in practice less than this due to movement and strain in the tissue), there is much less erosion, and the effect of applied load is reduced. In this regime, a wear mechanism may operate;Erosion still occurs even under a static applied load. Though the rate of erosion is smaller, it may be significant in vivo. The most likely mechanism, in this case, is a creep.

## Figures and Tables

**Figure 1 materials-14-00941-f001:**
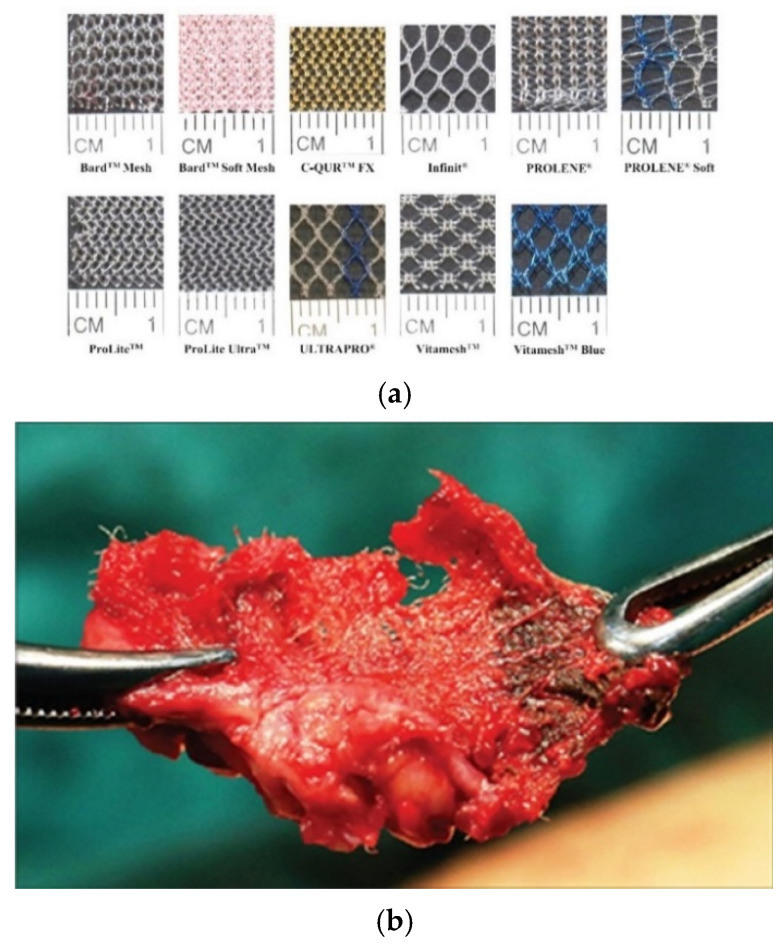
(**a**) Examples of different types of surgical mesh [1] and (**b**) a photograph of a piece of mesh, which was surgically removed after eroding through the wall of the patient’s bladder [2].

**Figure 2 materials-14-00941-f002:**
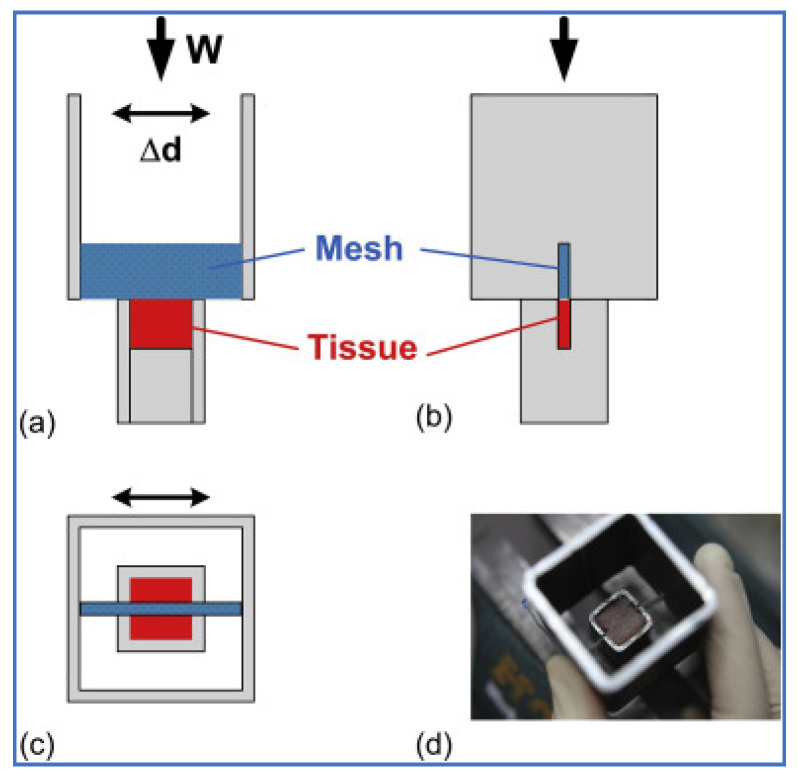
Original equipment developed for erosion testing by Taylor and Barton, 2020 [16]. (**a**) View from the front: a piece of mesh is secured in a metal holder, which slides back and forth a distance Δd while applying a force W. A sample of tissue is placed in a second holder and kept stationary. (**b**) View from the side: as the piece of the mesh erodes the tissue, it descends in two slots on the side of the holder, creating a vertical cut in the tissue. (**c**) View from the top. (**d**) A photograph looking down from the top after testing.

**Figure 3 materials-14-00941-f003:**
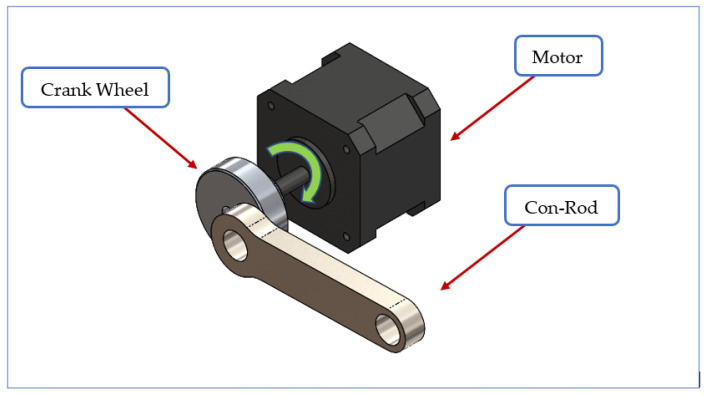
Motor and crank mechanism.

**Figure 4 materials-14-00941-f004:**
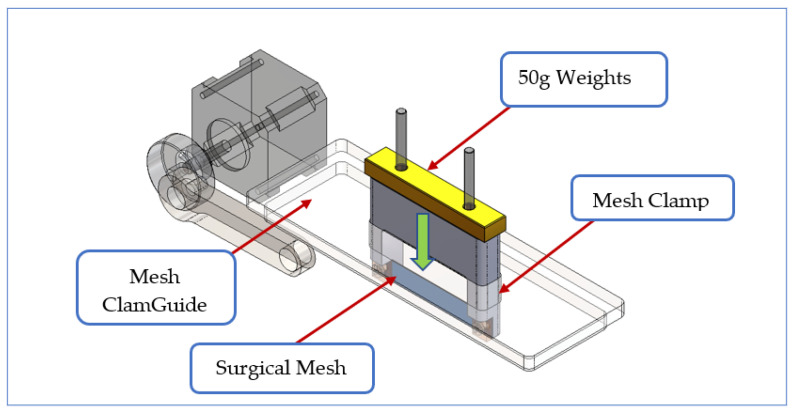
Mesh holder and guide to ensure free vertical movement.

**Figure 5 materials-14-00941-f005:**
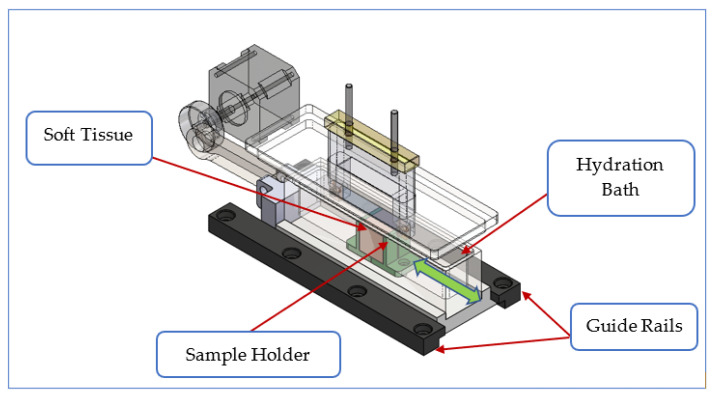
Soft tissue sample enclosed with X-axis only movement.

**Figure 6 materials-14-00941-f006:**
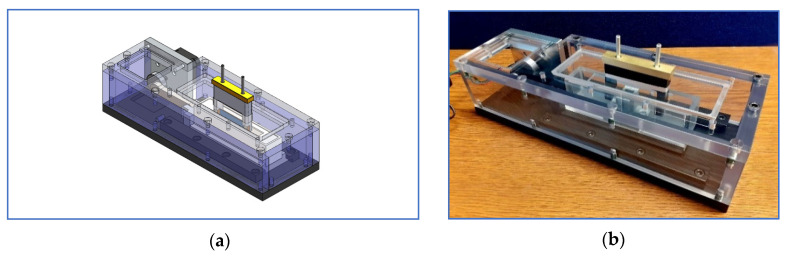
Fully enclosed testing machine: (**a**) CAD model and (**b**) manufactured machine.

**Figure 7 materials-14-00941-f007:**
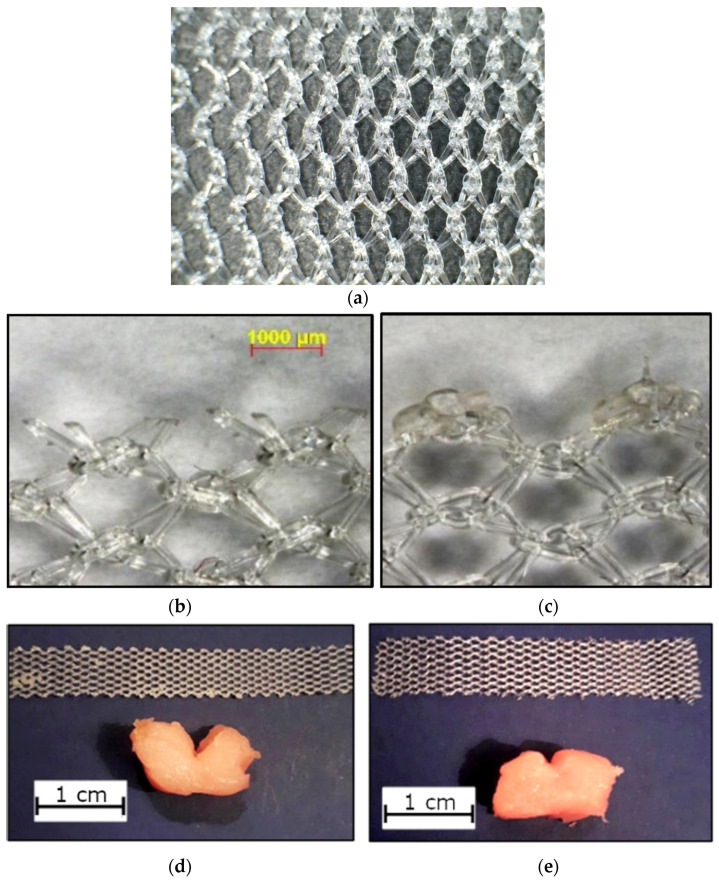
(**a**) Sutulene mesh material; (**b**) mesh edge cut with a scalpel; (**c**) mesh edge cut with a laser; (**d**) a specimen made with a laser-cut edge, showing the cut made in the tissue sample after testing; (**e**) as (**d**), but with a scalpel-cut sample tested under the same conditions. We used the depth of this cut to characterize the erosion rate.

**Figure 8 materials-14-00941-f008:**
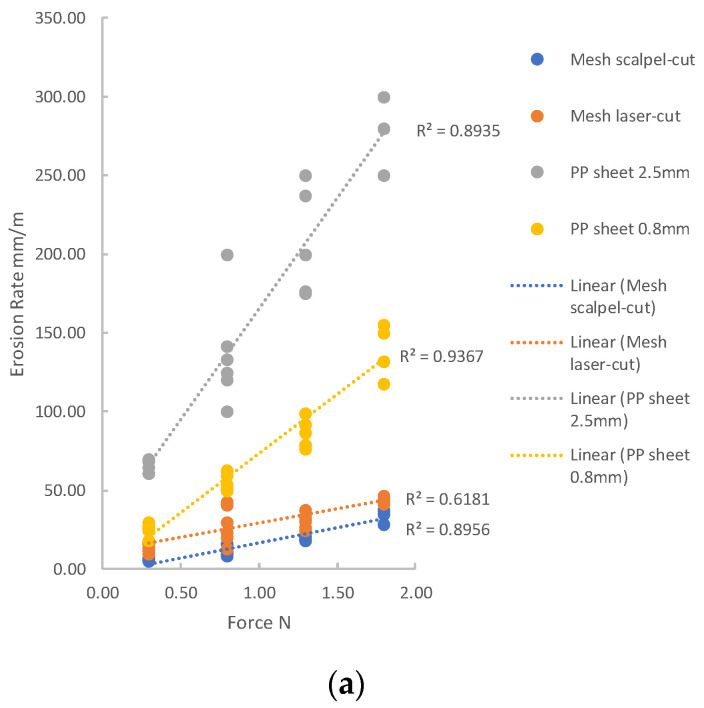
All results from tests using a stroke of 20 mm. Measured erosion rate as a function of applied force. (**a**) and (**b**) are two graphs with different vertical scales are used to best present the data. The lines indicate the results of best-fit linear approximations to the data, with accompanying R^2^ values.

**Figure 9 materials-14-00941-f009:**
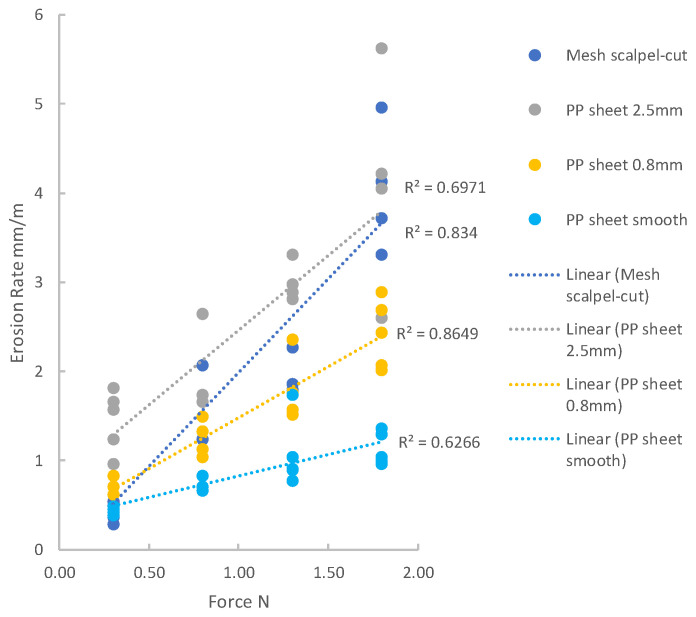
All results from tests using a stroke of 2 mm. Erosion rate as a function of applied force with linear fits and R^2^ values.

**Figure 10 materials-14-00941-f010:**
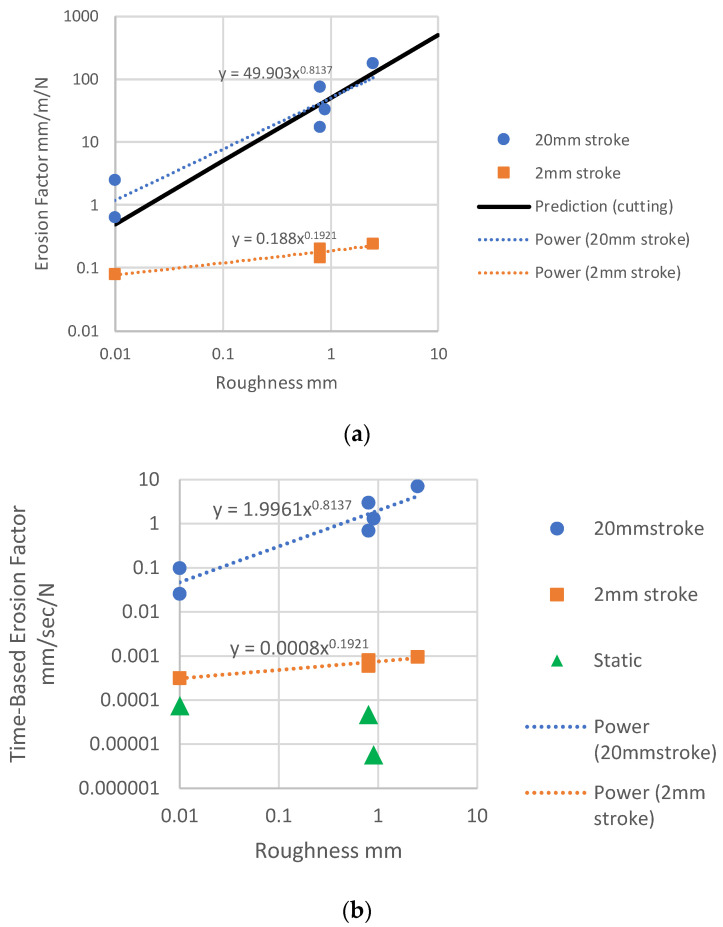
(**a**) Erosion factor as a function of roughness for the two different strokes. Power-law fitting lines through the data, and a prediction line using a simple model of cutting. (**b**) Time-based erosion factor, showing all groups tested.

**Table 1 materials-14-00941-t001:** Technical details of the test equipment.

Speed	Load	Stroke	Electrical Supply	Environment
0.01–10 Hz	0.3–3 N	1–40 mm	AC 240 V 50 Hz	Liquid bath: room temperature

**Table 2 materials-14-00941-t002:** Erosion rate (in mm/m) for tests conducted with a stroke length of 20 mm. Averages from at least 5 results per group: standard deviation in brackets.

Sample	Load = 0.3 N	Load = 0.8 N	Load = 1.3 N	Load = 1.8 N
Mesh scalpel cut R = 0.8 mm	6.25 (1.03)	11.29 (2.24)	19.28 (1.09)	34.86 (3.68)
Mesh laser cut R = 0.9 mm	13.70 (3.19)	29.44 (11.21)	30.58 (5.02)	44.08 (2.16)
PP sheet R = 2.5 mm	65.89 (3.66)	136.67 (34.08)	207.83 (34.53)	276.67 (25.17)
PP sheet R = 0.8 mm	25.34 (4.54)	55.36 (5.63)	86.47 (9.19)	140.83 (15.79)
PP sheet smooth	0.65 (0.17)	1.73 (0.12)	2.35 (0.36)	6.60 (1.08)
Suture	0.27 (0.08)	0.57 (0.13)	0.63 (0.09)	0.82 (0.32)

**Table 3 materials-14-00941-t003:** Erosion rate (in mm/m) for tests conducted with a stroke length of 2 mm. Averages from at least 5 results per group: standard deviation in brackets.

Sample	Load = 0.3 N	Load = 0.8 N	Load = 1.3 N	Load = 1.8 N
Mesh scalpel cut R = 0.8 mm	0.44 (0.11)	1.82 (0.37)	2.36 (0.56)	3.76 (0.86)
PP sheet R = 2.5 mm	1.45 (0.34)	1.87 (0.44)	2.98 (0.19)	3.88 (1.20)
PP sheet R = 0.8 mm	0.68 (0.17)	1.26 (0.18)	1.79 (0.33)	2.42 (0.38)
PP sheet smooth	0.42 (0.04)	0.77 (0.08)	1.07 (0.38)	1.23 (0.19)

**Table 4 materials-14-00941-t004:** Erosion rate (in mm/day) for tests conducted with static loading. Averages from at least 5 results per group: standard deviation in brackets.

Sample	Load = 0.8 N
Mesh scalpel cut R = 0.8 mm	3.23 (0.51)
Mesh laser cut R = 0.9 mm	0.40 (0.10)
PP sheet smooth	5.19 (0.30)

## Data Availability

Raw data (individual test results) can be made available on request to the authors.

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
