# Peer review of "The Development of Equipment to Measure Mesh Erosion of Soft Tissue"

_materials, 2021, doi:10.3390/ma14040941_

Round 1

Reviewer 1 Report

The authors designed and built an innovative test rig to study the effect of repetitive loading at the organ/mesh interface due to everyday life activities on tissue erosion in patients with implants. The manuscript potentially contributes to the field by providing novel insights about internal organs erosion in patients due to contact with implants made from surgical mesh. I would recommend this manuscript for publishing with Journal of Materials after revising it based following comments:

  1. The mesh is perpendicular to the tissue in the test rig. Why did you choose this configuration?
  2. Tissues are usually anisotropic materials. Is this considered in placing the specimens in the test rig? Should be clearly stated.
  3. The size of tissue specimen is an 8 mm cube. Why did you select this size of specimen? How do you justify the edge effect on the response of tissue?
  4. The tissue is held in a holder which is almost fixing the tissue. The mesh is clamped and is under tension. How do you justify the boundary condition of specimens (both tissue and mesh)? Do the adopted boundary conditions replicate in-vivo boundary condition?
  5. Figure 4: I believe the arrows are indicating wrong parts.
  6. How was the edge roughness measured? I would recommend using a standard roughness indicator.
  7. How was the erosion measured? The authors mentioned “erosion was quantified by measuring the length of the cut formed in the tissue sample” however it’s not clear if the “length of the cut” is the size of the opening or the depth of the cut? I recommend illustrating this visually in figure 7.
  8. Figure 7 shows that mesh edge cut with a scalpel has sharp edges while mesh edge cut with a laser has a rather dull. It is expected that sharp edges induce higher erosion rate however, the authors mentioned [line 209 page 8] “the mesh with a laser-cut edge had a significantly higher erosion rate than the mesh with a scalpel cut edge”, this should be discussed and clarified.
  9. I would recommend expanding the discussion by addressing findings from other researches on organs damage properties and comparing them with your findings.

Reviewer 2 Report

Abstract

The abstract is clearly presented.

Introduction

In the text, the references format does not match the manuscript template. Please review this throughout the manuscript.

The authors say “Despite the importance of mesh erosion, our search of the literature revealed no previous attempts to reproduce and measure the phenomenon in vitro. (line 62-63)” However, really to date (18/01/2021) your work in (Taylor and Barton, 2020), your work in (Schmidt and taylor, 2021) and the thesis of Amanda SCHMIDT (SCHMIDT, AMANDA, Characterization of the Erosion of Soft Tissue by Surgical Mesh In Vitro, Trinity College Dublin.School of Engineering, 2021) can already be considered a previous attempt. In other words, this paper is not the first work on the subject.

The reference “Schmidt and Taylor, Erosion of Soft Tissue by Polypropylene Mesh Products: JMBBM in press” in line 91 has not been included in the reference list. Please, include it.

Since the thesis of Amanda SCHMIDT provides important contents about the subject, it must also be cited.

Lines 88-97. Regarding the paper in (Schmidt and Taylor, Erosion of Soft Tissue by Polypro-91 pylene Mesh Products: JMBBM in press). Both, this and the mentioned paper, present experiments using the same automated machine and it is not clear in the introduction section why the present experiments and results are important to publish. Why were required the present experiments? the previous ones in (Schmidt and Taylor, 2021) were not enough? why the current experiments deserve another publication? Please clarify it in introduction section.

Methods and Materials

The experiments in this paper involve animal tissues. Please, provide ethic committee approval number and organization.

The authors present the manual machine version referencing Taylor & Barton (2020) in lines 98-117, and the author explain why an automated machine is required in Lines 114-115. So that the authors make It is clear that the experiments in (Taylor & Barton, 2020) were carried out using the manual machine version and that an automated machine was required for greater accuracy and longer experiments. However, one more time it is not clear why further experiments were required after the experiments in the paper in (Schmidt and Taylor, 2021). Please, explain it.

In subsection “moving tissue sample; sample hydration” in Line 146 the authors mention that the sample holder was 3D printed from ABS. How did you determine that ABS is an appropriate material for the "sample holder"? What kind of ABS is used? Is it an ABS biocompatible? The type of ABS used can get wet? Since the ABS absorbs moisture, the sample holder should must be replaced every certain period of time? The behaviour of the ABS with respect to humidity could alter the tissue sample? Please, explain it.

Line 120. what is the speed range that the machine allows? How to know the user how much speed he chose manually? please explain it.

Within the design parameters it is necessary to include the range of speeds required to satisfy the experiments that would normally be carried out using this machine.

Line 141. Why 0.3N force was selected? which was the required design requirement regarding this load?.  

Line 141. If additional weights can be added, how much is the weight limit? Please include weight requirements for this load within the design parameters.

Please, add a table or paragraph that summarizes the technical characteristics of the machine regarding speeds, loads, movement frequency in relation to speed motor, power supply, and additional technical characteristics that allow knowing the mechanical and electrical behavior of the machine.

Test Programme

The font on lines 161-167 looks different from the font on pages 168-172.

Line 161. what capabilities of the equipment? in terms of what? Please describe what capabilities of the machine you want to know through the experiments and describe why the current experiments are required regarding experiments in the paper in (Schmidt and Taylor, Erosion of Soft Tissue by Polypropylene Mesh Products: JMBBM in press).

Line 184. Why were loads of 0.3N, 0.8N, 1.3N and 1.8N selected? please explain what criteria you used to choose these loads.

Line 187. What criteria did you use to choose these testing times?

Lines 190-191. Please, add a reference to support how you calculate the erosion rate.

Results

Since results section presents statistical results, please add in “Methods and Material” section a paragraph that describes the methods used for statistical calculations. In addition, please include the software used for the calculations.

Discussion

I understand that the results in Figure 10 were obtained because it was considered convenient after analysing the results in Figures 8-9 and tables 1-3. However, erosion factor and time-based erosion factor as a function of edge roughness R should be addressed as part of your methods to analyse the data in “Methods and Materials” section. In addition, it would be convenient to move figure 10 to the results section so that the discussion contains only the "discussion" about erosion factor and time-based erosion factor and not the methods and results.

Conclusions

The conclusions are clear.

Round 2

Reviewer 1 Report

Thaks for the reponses and clarifications. It will be also benefitial if the authours address their future works like upgrading the test rig, as well.